# Risk of Periprosthetic Joint Infection after Intra-Articular Injection: Any Difference among Shoulder, Knee and Hip?

**DOI:** 10.3390/healthcare12111060

**Published:** 2024-05-23

**Authors:** Giovanni Vicenti, Federica Albano, Claudio Buono, Anna Claudia Passarelli, Elisa Pesare, Giulia Colasuonno, Teresa Ladogana, Biagio Moretti, Giuseppe Solarino

**Affiliations:** Orthopaedic & Trauma Unit, Department of Translational Biomedicine and Neuroscience (DiBraiN), School of Medicine, University of Bari Aldo Moro, AOU Consorziale “Policlinico”, 70124 Bari, Italy; dott.gvicenti@gmail.com (G.V.); f.albano34@studenti.uniba.it (F.A.); a.passarelli2@studenti.uniba.it (A.C.P.); e.pesare@studenti.uniba.it (E.P.); g.colasuonno10@studenti.uniba.it (G.C.); t.ladogana@studenti.uniba.it (T.L.); biagio.moretti@uniba.it (B.M.); giuseppe.solarino@uniba.it (G.S.)

**Keywords:** periprosthetic joint infection, joint disease, intra-articular injections

## Abstract

Osteoarthritis is a degenerative joint disease caused by the wear and tear of joint cartilage. The definitive and resolving treatment is prosthetic replacement of the articular surface, the demand of which is on the rise for patients with mild to moderate severity. However, a conservative strategy may be considered that aims to reduce and contain pain symptoms by postponing surgical treatment in the case of worsening that can no longer be otherwise controlled. Intra-articular infiltrations, like other therapeutic strategies, are not without complications, and among these the most feared is joint infection, especially in anticipation of future prosthetic replacement. Is important to avoid periprosthetic joint infections because they represent one of the third most common reasons for revision surgery. Using cases found in the literature, the aim of this article is to determine if there is a real correlation between the type of injections, the number of doses injected and the time between infiltrations and the surgical procedure.

## 1. Introduction

Osteoarthritis (OA) represents one of the main causes of disability, especially in the elderly. It has been estimated that the prevalence of this disease is set to increase in the coming years. Arthroplasty represents the main successful and prominent surgical procedure to treat OA of different joints in the late stage. Despite this, the implant of a prosthesis is associated with different types of complications. Meanwhile, intra-articular injection is the treatment choice in the early stage of OA, which is useful in controlling a patient’s symptoms. As shown in the literature, intra-articular injections before the implant of a prosthesis could be responsible for an increased risk of periprosthetic joint infection (PJI). PJIs represent one of the third most common reasons for revision surgery [1]. In the past, different studies have failed in the search for a direct correlation between an intra-articular injection before hip, knee and shoulder arthroplasty and the risk of PJI [2,3]. However, recent data show that the risk of infection seems to be correlated with the time between the injections and the surgical procedure and also to the number of injections before the implant of a prosthesis [4,5]. But these studies [2,3,5,6,7,8,9,10,11,12,13,14,15,16], due to their small sample sizes, were not able to find a real correlation between intra-articular injections and periprosthetic joint infections. The aim of this work is to provide an update on the literature by investigating the correlation between the type of injections, the number of doses administered and the time elapsed between infiltrations and the surgical procedure.

## 2. Materials and Methods

According to the Preferred Reporting Items for Systematic Reviews and Meta-analyses (PRISMA) guidelines [17], a comprehensive literature examination was conducted between January 2005 and January 2024 in the PubMed databases. We applied filters to include only English-language studies.

Three authors of this study collected literature separately (Albano F., Buono C. and Passarelli A.C.) and then screened abstracts and full texts.

Each paper was evaluated by one independent investigator (Vicenti G.); in cases of disagreement, it was solved by consensus with a third author with extensive expertise in implant-related infections (Solarino G.).

Levels II and III of evidence were evaluated for the following inclusion criteria: reviews, systematic reviews and meta-analyses focusing on infection in the orthopedic field and using as keywords “joint injection” and “periprosthetic joint infection” with the Boolean operators AND and OR.

We excluded studies centered on in vitro or in vivo animal models, papers with exclusive abstracts available, papers that made no mention of periprosthetic joint infections after total hip, total knee and shoulder arthroplasty and low-evidence research (expert opinion, technical comments and clinical trials).

Using a Microsoft Excel (Ver. 16.64) spreadsheet, some information was gathered following the authors’ protocol, including the study’s design, year of publication, first author and title (Figure 1).

## 3. Results

The preliminary search of the PubMed databases resulted in a total of 85 publications. Initially, 13 duplicates were eliminated. Afterward the analysis of the titles and abstracts, a total of 37 articles remained. We read the full text of the remaining 37 articles, and 21 were excluded because they did not meet the inclusion criteria. A total of 16 studies that met the inclusion criteria were selected. Despite the large number of studies selected, a limitation of our work could be the reliance on a single database, even though it is the most extensive and comprehensive one available in the literature.

## 4. Discussion

Osteoarthritis is a degenerative joint disease caused by the wear and tear of joint cartilage. The symptom spectrum includes slowly progressive worsening of joint function, pain and eventual deformity of the anatomical profile [18]. Although the definitive and resolving treatment is prosthetic replacement of the articular surface, the demand of which is on the rise, for patients with mild to moderate severity, however, a conservative strategy may be considered that aims to reduce and contain pain symptoms by postponing surgical treatment in the case of worsening that can no longer be otherwise controlled [19]. The conservative approach considers lifestyle changes, weight loss, pharmacological and physiotherapeutic analgesic therapy and intra-articular joint injections [18].

These injections can be performed with corticosteroids and local anesthetics, hyaluronic acid (HA) or platelet-rich plasma (PRP) [20,21,22]. Among them, corticosteroids interrupt the inflammatory reaction and immune response and reduce cartilage erosion and osteophyte formation [2]. On the other hand, a high dose of corticosteroids could be responsible for cartilage lesions, because the corticosteroids could be trapped in peri-articular soft tissues or in the degeneration areas of the joints [2]. The persistence of corticosteroids in deep tissue may be responsible for local immunosuppression after joint arthroplasty [1]. Despite these effects, different studies have been published about the effect of corticosteroid injections (CSIs) in the treatment of symptomatic OA [23]. A recent randomized trial compared the intra-articular effect of a corticosteroid (triamcinolone) with the effect of a placebo [23]. Even though patients in both groups reported a reduction in pain, those treated with triamcinolone experienced symptom improvement just one week after the first injection, whereas the placebo-treated group showed pain relief only after the third injection [23]. Intra-articular infiltrations, however, like other therapeutic strategies, are not without complications, and among these the most feared is joint infection, especially in anticipation of future prosthetic replacement [7]. PJIs represent one of the third most common reasons for revision surgery [1]. In the literature, it has been reported that the incidence of PJI ranges between 0.5% and 2.83%, and these values are set to increase in coming years [1]. In the recent literature, it is not clear which component of the injected drugs or what phase of the injection procedure could be responsible for the increase in infection risk [2]. One of the main causes could be the direct inoculation of the joint or the local immunosuppression effect of some of the drugs injected [3]. During the injections, a small amount of bacteria could be inoculated in the joint space, and later they could colonize the area [3].

Moreover, the optimal timing between the last intra-articular injection and surgery remains uncertain. Determining a precise and safe interval between these two procedures is crucial to minimize the risk of PJI [7].

The purpose of this study is to offer an updated review of the literature by examining the relationship between the type of injections, the number of doses administered and the time interval between the infiltrations and the subsequent surgical procedure.

### 4.1. Hip Infiltrative Injections and PJIs

Corticosteroid injections have emerged as a crucial strategy in pain management, particularly in patients who are delaying total hip arthroplasty (THA) [24]. In recent times, there has been rising apprehension regarding the potential elevation of postoperative infection risk associated with preoperative CSIs [18]. Nevertheless, despite these concerns, the existing body of research has not definitively determined the safety of CSIs when administered before THA [24,25].

Forlenza et al. [6] carried out a study from 2011 to 2018. A pool of 29,058 patients who underwent hip steroid intra-articular infiltration in the 6 months preceding THA was analyzed. The number of injections was also evaluated. At a 6-month follow-up, the incidence of periprosthetic joint infections (PJIs) was 1.79% [6]. A temporal and dose-dependent correlation was demonstrated. CSIs administered in the 4 months preceding surgery exhibited a significant association with periprosthetic joint infections (PJIs), with a rate of 1.6%. Additionally, within 4 months, the incidence of PJIs demonstrated an escalating trend: 3 months, 1.9%; 2 months, 1.9%; and 1 month, 2.6%, with *p* value < 0.05 [6]. Conversely, injections administered more than 4 months before THA did not show an increased risk of PJI (*p* > 0.05). Regarding the dose-dependent relationship, the risk of PJI doubled with two injections, and when more than three injections were administered, the odds of developing a PJI increased 3.5-fold [6].

Kaspar et al. [9], in a cohort study, did not find a specific relationship between the preoperative number injections of corticosteroids and the postoperative risk of periprosthetic joint infection after total hip arthroplasty. The authors conducted a retrospective study collecting data derived from two groups: a group of patients who had received CSIs before surgery and a second one who had not received intra-articular injections [9]. The results showed that the first group had a higher risk of sepsis of the hip (4 times higher) than the second group [9]. In addition, the revision rate for the first group was 12.5%, in contrast with the second group (1.02%) [9]. Kaspar and colleagues tried to establish a direct correlation between injections and the procedure of the injections and PJIs, but the small number of patients taken into account (only 40 patients enrolled) did not allow them to find significant evidence of correlations between infections and injections [9]. The authors suggested at the end of their work to avoid intra-articular injections in patients eligible for THA [9].

Chambers et al. [8], in a cohort study published in 2017, demonstrated a higher risk of infection after two preoperative injections of corticosteroids than a single dose. After a double preoperative injection, patients showed an infection rate that was triple that compared to a single preoperative dose [8]. The results of this study were similar to those of Forlenza et al., but in these cases, the small sample size also could represent a limit for these studies [8].

A metanalysis carried out by Saracco et al. [7] demonstrated a statistically significant 20% increased risk for PJI for patients who underwent intra-articular injections prior to THA. This work analyzed data derived from 308,810 patients and showed an increased risk of PJI in the injection group [7]. Concern the timing of the injections, Saracco and colleagues found out that the risk of PJI is higher (64%) when the injection is performed 3 months before THA, while for the interval of 6–12 months, no clear data have emerged [7]. In addition, Saracco et al. [7] analyzed the role of corticosteroids versus HA in the occurrence of periprosthetic infections. It appears that prolonged immunosuppression caused by CSs underlies the pathogenesis of PJIs.

Although some authors have pointed out that the interval between infiltration and THA may influence the occurrence of periprosthetic infections, Li H. et al. emphasized that there are no clear guidelines for timing management due to the paucity of data in the literature [10].

Albanese et al. [11] conducted a qualitative analysis of the literature on patients undergoing CSIs before THA. The authors performed a meta-analysis, collecting data from 198,997 patients [11]. Of these, 54,355 patients received CSIs after or before 3 months from THA [11]. In addition, injection timing took a relevant role, with a higher risk if infiltrations were performed within three months before total hip arthroplasty [11]. What has emerged is a statistically significant association between PJIs and CSIs [11]. Another important piece of information obtained from this study is that CSIs are responsible for an increase in the risk of infection in the THA procedure (1.4% compared to 0.8% in patients who did not receive CSIs before surgery), while an increase risk in TKA is not confirmed yet [11]. This difference between THA and TKA is not easy to explain. Albanese tried to explain this aspect with two theories: the first one is the longer operative time of THA against TKA; the second one is that bacteria have different resistance patterns depending on the site (knee or hip) [11]. Despite these theories, the recent literature seems to be insufficient. Another aspect of this study is related to the timing of the injections before surgery [11]. The risk of infection was higher in patients who received injections < 3 months before surgery compared to patients injected >3 month before THA [11]. However, this study shows different limitations, such as the fact that for the patients enrolled the injection protocol was not standardized, and not all the retrospective studies analyzed reported the timing of injection [11]. 

Lai et al., in 2022, published an article in which they compared data from 173,465 arthroplasties in hips and in knees with data from a group of 100,416 control patients. The infection risk in patients who had received intra-articular injections of corticosteroids in the hip 3 months before the surgical procedure was higher compared with patients who did not receive injections [1]. Moreover, no differences in the infection risk were found in patients who had been injected between 3 and 6 months or between 6 and 12 months before arthroplasty [1]. However, this study has several limitations [1]. For example, the data collected coming from different databases also included data from studies with a small sample size [1]. Another limitation was the lack of clear data about timing of the injection, and this is the reason why the authors used a large time window for this work (0–3, 3–6 and 6–12 months) [1]. In conclusion, Lai et al. and other authors recommend avoiding intra-articular injections within 3 months of the surgical procedure [1]. Other studies are mandatory to confirm these findings.

### 4.2. Knee Infiltrative Injections and PJI

Intra-articular injections of corticosteroids and hyaluronic acid (HA) are responsible for pain relief in the early stage of osteoarthritis, while total knee arthroplasty (TKA) is required in the final stage [26]. About 30 percent of patients undergoing TKA received at least one infiltration before their surgical procedure [26].

In addition to THA, the role of intra-articular injections in increasing the risk of PJI before TKA is controversial. Data extracted from the recent literature underline the lack of evidence in establishing a direct correlation between injections and PJIs [4].

Baums et al. pointed out a positive correlation between TKA infection and CSIs [4]. These authors conducted a systematic review collecting retrospective studies from different databases (PubMed, Cochrane and Scopus) [4]. From the six retrospective studies analyzed in detail, Baums et al. identified 255,627 patients who underwent TKA [4]. Of these, only 80,579 patients received intra-articular injections [4]. The result of this study was that CSIs have been related to a higher risk of PJI [4]. The authors also underlined that the reasons for the higher risk of PJI after CSIs could be two: the first reason could be the possibility that bacteria from skin flora may be inoculated into the joint and activated during the surgical procedure; the second reason is represented by the immunosuppressive effect of corticosteroids, which could inactivate host defenses [4]. These two reasons combined may explain why other injections such as PRP or HA are characterized by a lower risk of PJI than CSIs [4]. However, the main limitation of this study was that the articles selected did not evaluate the time spent between injections and surgery [4].

Bhattacharjee et al. [13] analyzed TKA patients who received CSIs within 6 months before surgery. Data were extrapolated from a national database from 2007 to 2017 [13]. It was observed that injections performed within 2 weeks before surgery had an increased risk of postoperative PJI (*p* = 0.02). Comparable outcomes were found even in case of infiltration executed within 2–4 weeks before TKA [13].

Yang et al. [12] in 2021 conducted a meta-analysis comparing the results of eight articles published between 2005 and 2008. This study showed that the infection rate of patients who received injections before TKA (73,880 patients) was higher than that of the control group (200,067 patients who did not receive any injections before surgery) [12]. Yang et al. [12] also underlined that the mechanism with which injections could lead to PJI is unclear. The most reliable theory is that the injections could deliver bacteria into the joint cavity even if an aseptic technique is used [12]. Another theory is that corticosteroids could cause a local immunosuppression that causes an increase in the infection rate [12]. HA, as shown in the literature, may act as an immunosuppressor agent, altering the production of immunomodulatory factors in the joint cavity [12]. Yang et al. [12] concluded their study by showing that there was no difference in the infection rates between patients who received HA joint injections and patients treated with intra-articular CSIs [12].

While different studies in the literature have been published about the higher risk of PJI after CSIs, few works have evaluated the risk of infection after HA injections.

In 2019, Richardson and colleagues published a work in which they presented a comparison of the infection risk of PJIs between patients who received CSIs and who received HA before TKA [3]. They collected data from 58,337 patients selected from a national database. Each patient received intra-articular HA injections or CSIs ≤ 1 year before surgery [3]. The results were compared to those deriving from a control group of non-injected patients [3]. A total of 1649 cases of infections were detected, and of these, 1052 cases were diagnosed in the control group [3]. Regarding the injected patients, the infection risk was 53% for those who received intra-articular HA infiltrations within 3 months before TKA, compared to 19% for those who received CSIs within the same timeframe before the surgical procedure [3]. In patients who received intra-articular infiltration >3 months prior to TKA, no statistically significant increase in the infection rate was detected [3]. The reason why HA injections could be responsible for an augmented risk of PJIs is not clear. Richardson et al. [3] said that, as shown in different experiments found in the literature, HA could reduce the immune response by altering the production of immunomodulating factors in cartilage, synovium and other peri-articular tissues. It is possible that a combination of direct inoculation and the local immunosuppressor effect could increase the risk of PJI [3]. In conclusion, Richardson et al. found no significant differences in PJI risk between different drugs injected or between patients who received multiple injections vs. a single injection [3]. In conclusion, the authors recommend avoiding any type of injection ≤3 months prior to the surgery procedure [3].

In 2022, Rodriguez et al. published a retrospective study about the risk of PJI in patients who received CSIs before TKA [14]. In their work, a total of 47,903 patients had been recruited from a national database [14]. Each patient underwent a primary unicompartmental knee arthroplasty (UKA) after at least one ipsilateral CSI [14]. Few works in the literature have been published concerning the risk of PJI after intra-articular joint infection and partial knee arthroplasties. Rodriguez and colleagues, in their study, found that the risk of PJI after CSIs administered 6 months before surgery was higher (nearly 2.1% versus 1.3% of the) at 6 months, 1 year and 2 years after the injection compared to the same risk in the control group composed of patients not subjected to intra-articular injections (nearly 2.1% versus 1.3%) [14]. A large portion of the cases of PJI occurred in the first 6 months after the injection (75%), while no differences in the risk of infection were found from 6 months to 1 year [14]. But this work has several limitations, and one of them is that it is not possible to define if the cause of the infections reported is specific bacteria or if the causes are multifactorial [14]. Another limitation is that, due to the lack of data from this database, no correlation could be made between infection and the timing and number of intra-articular injections before surgery [14]. Because the use of UKA has grown in recent years, the risk of PJI after this type of surgical procedure is gaining more interest [14]. Therefore, despite its limitations, the work of Rodriguez et al. is one of the most important studies regarding the importance of avoiding PJIs in UKAs [14].

### 4.3. Shoulder Infiltrative Injections and PJI

The incidence of PJI following shoulder arthroplasty is around 4%, and these infections represent the second reason for revision in this surgical procedure [5]. Intra-articular injections of corticosteroids seem to be responsible for an increased risk of PJI in shoulder arthroplasty, even though, as already highlighted, there is still a lack of evidence in the literature [16]. Corticosteroids or other drugs could be injected into the glenohumeral joint through an anterior or posterior approach, into tendon sheaths, into tender points or into the subacromial space [2]. Moreover, each patient could be treated with single or multiple injections, injections in different sites at the same time, different corticosteroid formulations, different volumes injected and different types of anesthetics used [2].

In 2015, Rashid et al. [2] undertook a retrospective study of patients undergoing shoulder arthroplasty after having received intra-articular joint injections. The aim of the study was to underline the correlation between injections and periprosthetic joint infection [2]. The authors enrolled 83 patients divided into two groups: 23 patients who received CSIs approximately 11.4 months before their surgical procedures and 60 patients who did not receive any injections before surgery [2]. Of the first group, none of the patients developed a superficial surgical site infection, while just one had the development of deep surgical site infection, which led him to a prosthetic revision [2]. However, no clear evidence of any correlation between PJI and injections was shown, which contraindicates the latter, probably due to the limitations of the study [2].

Werner et al. [15], in 2015, carried out a multicentric study based on a national database to analyze the complications of injections in patients undergoing shoulder arthroscopy and arthroplasty. The sample was divided into six cohorts of patients (three groups for arthroscopy and three groups for arthroplasty, respectively) [15]. Intra-articular corticosteroid administrations at 3 months and between 3 and 12 months before surgery were evaluated [15]. Regarding prosthetic replacement, the incidence of periprosthetic infection in the case of infiltration performed 3 months before surgery was significantly higher [15].

Baksh et al. [5], in 2023, published two different studies. The first one was a cohort study in which the enrolled patients were divided into four groups (patients who received CSIs less than one month before, patients who received injections one to two months before, patients who received injections two to three months before and patients who received CSIs after three months before shoulder arthroplasty surgery) [5]. The authors found a time-dependent relationship between CSIs and infections [5]. In fact, this study demonstrated that CSIs administered one month before surgery increase the risk of PJI [5]. But, as other in studies, this one has the same limitations, which are the sample size and data from patients followed by the same institution [5].

In March 2023, Baksh et al. [16] published a second work creating a national database of patients to demonstrate a correlation between CSIs and PJI following shoulder arthroplasty. The patients analyzed were divided into four groups: 214 patients who underwent shoulder arthroplasty procedure 4 within 4 weeks after CSIs, 473 who were subjected to the same surgical procedure after 4–8 weeks from CSIs, 604 patients who received CSIs 8–12 weeks prior to the surgical procedure and a control group [16]. The data of the first three groups were compared to the data collected from the control group who did not receive CSIs before the surgical procedure [16]. This study demonstrated that CSI timing is linked to the risk of PJI, especially when injections were performed 4 weeks prior to the surgical procedure [16]. Although these results appear to be nothing new compared to the recent literature, Baksh et al., unlike the authors of other studies, suggested specifically that a shoulder arthroplasty should be performed at least 4 weeks after CSIs in order to guarantee a lower risk of PJI [16].

## 5. Conclusions

Although corticosteroid and HA infiltrations represent a valid, minimally invasive therapeutic strategy in the treatment of osteoarthritis of the hip, knee and shoulder, particular attention must be paid to the time interval between the administrations themselves and the definitive prosthetic replacement surgery. Data extrapolated from the literature, although sometimes limited by the paucity of samples in the studies, show an increased risk of periprosthetic infections in patients undergoing cycles of infiltration in the preoperative setting. This is both dose-dependent and time-dependent and increases in the case of multiple administrations and if the interval between the infiltration itself and surgery is less than 3 months. The literature is particularly prolific on hip and knee infections, while appearing significantly less comprehensive on the upper extremity. This aspect could be a springboard toward attempting to broaden the landscape of knowledge so as to establish clear guidelines that will allow for the appropriate management of this problem in the future.

## Figures and Tables

**Figure 1 healthcare-12-01060-f001:**
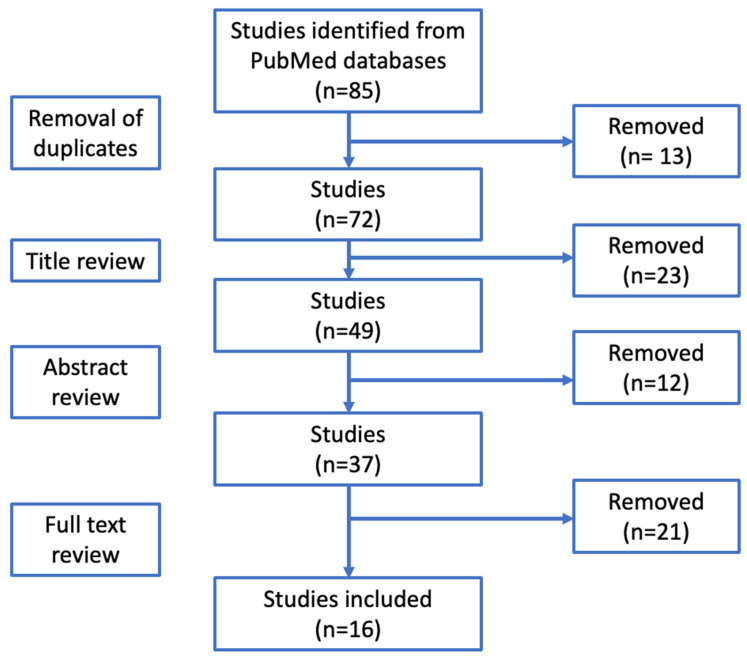
PRISMA flow diagram.

## Data Availability

Not applicable.

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
