# Peer review of "Risk of Periprosthetic Joint Infection after Intra-Articular Injection: Any Difference among Shoulder, Knee and Hip?"

_healthcare, 2024, doi:10.3390/healthcare12111060_

Round 1

Reviewer 1 Report

Comments and Suggestions for Authors

A review article by Vicenti et al. discusses the use of injections in the conservative treatment of osteoarthritis. It discusses how osteoarthritis leads to the need for prosthetic replacement of the affected joint surface. However, conservative methods, including delivery injections, are increasingly being used to relieve pain and delay surgery, especially in patients with mild to moderate disease severity.

This paper discusses the potential risks and benefits of using delivery injections, with a particular focus on the risk of periarticular joint infection, which can complicate future prosthetic replacement and require revision surgery. Existing literature has been reviewed to determine if there is an association between the type, frequency, and timing of delivery injections and the likelihood of subsequent surgery. The topic addressed is very timely and relevant to improving the quality of life for patients facing these issues.

In my opinion, the structure of the article is clear, the methodology is correct, and the literature cited is current. 

Author Response

Dear reviewer,

We appreciate you for you precious time in reviewing our paper and providing valuable suggestions. We greatly appreciate the feedback given on our work and "and we would be honored to publish our paper in your journal.

Reviewer 2 Report

Comments and Suggestions for Authors

The introduction section is insufficient for publication in this journal, so changes should be made. The authors refer to various studies without any citation and then discuss cases found in the literature without citation, based on which the purpose of the study was formulated.

1.      As shown in literature intraarticular injections before an implant of a prosthesis could be responsible for an increased risk of periprosthetic joint infection (PJI). Please kindly cite the references in the literature.

2.      In the past, different studies have failed in the search for a direct correlation between intra-articular injection before hip, knee and shoulder arthroplasty and the risk of PJI. Which are those studies you refer to?

3.      However recent data show that the risk of infection seems to be correlated to the time between the injections and the surgical procedure and also to the number of injections before the implant of a prosthesis. Where is that recent data found?

4.      But these studies, cause the small sample sizes, are not able to find a real correlation to intra articular injections and periprosthetic joint infections. Using the cases found in literature, the aim of this article is to determine if there is a real correlation between the type of injections, the number of doses injected and the time between infiltrations and the surgical procedure. Which are the studies and cases you refer to?

5.      At the end of the introduction, please specify exactly what is novel about this study and its place in the scientific literature.

6.      The introduction needs to be almost completely rewritten, with bibliographical references.

7.      As I understand it, the authors only used a single database, which is quite insufficient for a proper systematic review. This should be specified as a limitation of the study.

8.      Specifying which keywords were used to find the articles in the PubMed database is mandatory.

9.      The Materials and Methods should specify exactly what the typology of the study is, which it is strongly recommended to be included in the title.

10.   The first sentence of the discussion should relate to the study's objectives. Have they been achieved or not?

11.   Reference number 9 does not appear chronologically/adequately in the text. Please fix the problem.

12.   The authors also aimed to extract articles according to the year of publication of the studies? If yes, then it should be specified.

Comments on the Quality of English Language

Moderate editing of the English language is needed.

Reviewer 3 Report

Comments and Suggestions for Authors

In this manuscript, the authors analyzed literature data to check for correlations between the timing of intra-articular injection and the frequency of infections following hip, knee, and shoulder arthroplasty. The authors included papers published betweem January 2005 and January 2024 in the Pubmed databases. After applying the inclusion and exclusion criteria, the list of papers was narrowd down to 16 studies. For hip and knee arthroplasty, the aggregate literature data indicates that intra-articular corticosteroid injections (especially within 3 months of surgery) increase the risk of post-surgical infection. The literature data is less comprehensive for the shoulder, but the trends are similar to the hip and knee.

I have the following questions and comments for the authors:

(1.) Please include proper citations in the Introduction section. For example, please include the citations for the following statements: (a) “As shown in literature intraarticular injections before an implant of a prosthesis could be responsible for an increased risk of periprosthetic joint infection (PJI).” (lines 28 to 30, page 1) (b) “PJIs represent one of the third most common reason for revision surgery.” (lines 30 and 31, page 1) (c) “In the past, different studies have failed in the search for a direct correlation between intra-articular injection before hip, knee and shoulder arthroplasty and the risk of PJI. However recent data show that the risk of infection seems to be correlated to the time between the injections and the surgical procedure and also to the number of injections before the implant of a prosthesis.” (lines 31 to 35, page 1)

(2.) In lines 42 and 43 of page 1 (Materials and Methods section) the authors mention that they did a literature search according to the PRISMA guidelines. Please include a citation for these guidelines.

(3.) In lines 56 and 57, the authors mention that they excluded clinical trials because these are “low evidence research”. Please could you elaborate why you consider clinical trials to be low evidence research, and why you excluded the data from them?

(4.) In line 86 on page 3, the authors use the abbreviation “CSI” for the first time but there is no explanation for what this is. Please include the expanded form of each abbreviation (“corticosteroid injection” in this case) the first time you use it.

(5.) It is not clear what this statement (from lines 88 to 90 on page 3) means: “Even if patients of both groups reported reduction in pain, in the group of patients treated with triamcinolone the improvement of symptoms had been referred 1 week after the first injection [8].” What do you mean by “referred 1 week after the first injection”? Did the treatment group have an improvement after 1 week but the control group did not? Please explain more clearly.

(6.) What does the abbreviation “THP” (page 4, line 145) mean?

(7.) In several places in the manuscript, the authors use a dot while using larger numbers (for example, 198.977 on line 155, page 4). This might lead to confusion as a comma is more commonly used in the US, and the dot might be confused as being a decimal point. At other places, the authors do not use either a dot or a comma (for example, 54355 on line 155, page 4). Please use a consistent notation and please clarify the meaning to the readers if you use a dot or a comma.

(8.) On lines 218 to 220 on page 5, the authors mention “HA as shown in literature may act as an immunosuppressor agent altering the production of immunomodulatory factors in the joint cavity.”. Please include literature citation(s) for this.

(9.) It is difficult to understand what the authors mean in lines 231 and 232 of page 6: “For what concerned the injected patients, the infection risk was 53% in the patients who received intraarticular infiltrations of HA ≤ 3 months before TKA than patients who received CSIs (19%) with the same timing [10].” Please rewrite this in a clear manner.

(10.) The following sentence (lines 250 to 252, page 6) is also hard to understand: “Rodriguez et colleagues in their study had found a risk of PJI, after CSIs sub ministered 6 months before surgery, at 6 months,1 and 2years after the injection higher than the risk in the control group (nearly 2.1% against 1.3% in patients not injected) [22].” Please rewrite this in a clear manner.

(11.) In lines 297 and 298 of page 7, the authors mention “But as other studies, this one has several limitations such as the sample size and it is also limited to one institution [23].”. Are there any other limitations in this study? If yes, please include them here. Also what are the limitations in the other studies?

Comments on the Quality of English Language

Parts of the manuscript were hard to understand and/or grammatically incorrect. Manuscript needs significant editing for quality of English.

Round 2

Reviewer 2 Report

Comments and Suggestions for Authors

Congratulations on your hard work!

Comments on the Quality of English Language

Moderate editing of the English language is needed.

Reviewer 3 Report

Comments and Suggestions for Authors

The authors have addressed the comments from the previous round of review. I have no further questions or comments.